# A Reconfigurable Visual–Inertial Odometry Accelerated Core with High Area and Energy Efficiency for Autonomous Mobile Robots

**DOI:** 10.3390/s22197669

**Published:** 2022-10-09

**Authors:** Yonghao Tan, Mengying Sun, Huanshihong Deng, Haihan Wu, Minghao Zhou, Yifei Chen, Zhuo Yu, Qinghan Zeng, Ping Li, Lei Chen, Fengwei An

**Affiliations:** 1School of Microelectronics, Southern University of Science and Technology, Shenzhen 518055, China; 2Scientific and Technical Center for Innovation, Beijing 100080, China; 3Department of Computing and School of Design, The Hong Kong Polytechnic University, Hong Kong, China

**Keywords:** SLAM, VIO, accelerator, reconfigurable, AMRs

## Abstract

With the wide application of autonomous mobile robots (AMRs), the visual inertial odometer (VIO) system that realizes the positioning function through the integration of a camera and inertial measurement unit (IMU) has developed rapidly, but it is still limited by the high complexity of the algorithm, the long development cycle of the dedicated accelerator, and the low power supply capacity of AMRs. This work designs a reconfigurable accelerated core that supports different VIO algorithms and has high area and energy efficiency, precision, and speed processing characteristics. Experimental results show that the loss of accuracy of the proposed accelerator is negligible on the most authoritative dataset. The on-chip memory usage of 70 KB is at least 10× smaller than the state-of-the-art works. Thus, the FPGA implementation’s hardware-resource consumption, power dissipation, and synthesis in the 28 nm CMOS outperform the previous works with the same platform.

## 1. Introduction

In recent years, AMRs have achieved rapid development driven by practical application demands. As the core technology of AMRs, simultaneous localization and mapping (SLAM) include five parts as Figure 1 shows: sensor data reading, front-end visual odometry, backend optimization, loop closure, and mapping. As a single sensor cannot cope with all situations, the most effective approach is to fuse data from both the camera and IMU, an algorithm called VIO. Aside from the computational complexity of loop closure and global optimization, the VIO system can estimate the position and trajectory of a moving object in real time with lower power consumption, which is very important for lightweight AMRs with high processing speed requirements.

The visual front-end of VIO mainly estimates the robot’s motion through the direct method or feature point extraction. In 2008, the direct method proposed by Silveira G. et al. [1] used the difference in light intensity of each pixel in adjacent frame images to estimate the camera’s motion. At present, representative open-source projects are DTAM [2], LSD-SLAM [3], and DSO [4], etc. Compared with feature point extraction, the direct method saves time for calculating features and can maintain certain robustness when the texture is scarce, but it is difficult to work in scenes with drastic changes in light. Feature point extraction uses image features instead of image intensity, which overcomes the shortcomings of the direct method but relies on the feature extraction results. In recent years, classic features have included SIFT [5], SURF [6], FAST [7], and so on.

Backend optimization is to optimize the motion pose estimated by the front end to minimize accumulated errors. The main backend algorithms are divided into filtering methods using extended Kalman filter (EKF) or other filters and optimization methods using bundle adjustment (BA) or graph optimization methods. In the case of limited computing resources and relatively simple quantities to be estimated, the filtering method represented by EKF is very effective; however, because the storage capacity and state quantity are in a quadratic growth relationship, there are many feature data, and the filtering method is less efficient. In addition to KF [8] and EKF [9], there are also information filters [10,11] and particle filters [12,13,14]. Contrary to the filtering method, the optimization method no longer relies on the information at a specific moment but obtains the optimal global estimation of all landmarks by optimizing the joint error function of all poses and landmarks. Existing optimization methods include that by Di, K. et al. [15], who studied an extended BA algorithm that can utilize both 2D and 3D information and is suitable for RGB-D cameras. Alismail, H. et al. [16] abandoned the minimization of reprojection error and achieved a significant improvement in accuracy with a photometric BA algorithm based on maximizing photometric continuity. The incremental, consistent, and efficient BA proposed by Liu, H. et al. [17] adopts incremental technology, which consumes only about 1/10 of the computing resources of traditional BA under the premise of ensuring accuracy.

In addition to improving algorithms, using SLAM systems also relies on hardware acceleration. Compared with CPU, GPU has powerful floating-point computing capability, so GPU is generally used in the early stage of hardware acceleration design research, such as feature extraction accelerator based on SIFT [18] and SURF [19] features. However, the GPU system consumes a lot of power. With the development of field programmable gate arrays (FPGAs), hardware acceleration relies more on FPGA. Yum, J. et al. [20] designed a complete SIFT hardware accelerator, Wilson, C. et al. [21] proposed a complete FPGA implementation of the SURF accelerator, and Ulusel, O. [22] showed that feature extraction on the FPGA platform has great advantages over CPU and GPU.

Moreover, accelerators suitable for backend optimization have also been proposed recently. For example, Tertei, D. T. R. et al. [23] proposed an efficient FPGA SoC hardware structure using systolic array matrix multiplication to accelerate the EKF-SLAM algorithm. Wang, J. et al. [24] designed a reconfigurable matrix multiplication coprocessor for accelerating matrix multiplication in visual navigation algorithms. However, the hardware accelerators can only accelerate specific algorithms, or they are incomplete, only accelerating the feature extraction or matrix multiplication part and lacking pose estimation and trajectory output.

Currently, the relatively complete acceleration design includes Liu, R. et al. [25] implementing a complete ORB-SLAM system on FPGA. The feature extraction and feature matching parts are accelerated by FPGA, but the trajectory estimation and pose optimization are completed by ARM. Li, Z. et al. [26] presented an accurate, low-power, real-time CNN-SLAM processor that implements full-visual SLAM on a single chip. Suleiman, A. et al. designed a chip that can perform VIO [27], reducing on-chip storage to 1/4 the size through image compression techniques. Zhang, Z. et al. designed a VIO system that uses Kintex-7 XC7K355T (an FPGA) [28], and Wang, C. et al. designed a visual SLAM system that uses UltraScale+ XCZU7EV (an FPGA) [29].

This paper proposes a reconfigurable, real-time, low-area, and energy-efficient VIO accelerator implemented on an FPGA with the following major design features: (i) a reconfigurable accelerator architecture that adapts to different Kalman-filter-based VIO algorithms; (ii) an optimized instruction-based structure supporting the simultaneous workflow of fixed-point and floating-point units to accelerate VIO algorithms for real-time usage. This design can support the post estimation and trajectory output of real-time > 60 Hz frame input and > 200 Hz of IMU input; and (iii) a computing core with shared memory and the memory reuse strategy. This works only consumes 70 KB of on-chip memory, which is at least 10× lower than the previous works. To our knowledge, this is the first integrated reconfigurable architecture that supports multiple VIO algorithms implemented in FPGA.

## 2. Algorithms

### 2.1. Overall Procedure of the Visual–Inertial Odometry

Figure 2 shows the overall procedure of the visual–inertial odometry, consisting of IMU pre-integration, feature coordinates optimization, and extended Kalman filter (EKF). IMU pre-integration computes the prior estimate of pose and the position and other state variables, such as velocity and feature location of the AMR concerning the IMU. Feature coordinates optimization optimizes the pre-estimated pixel location of the stored features by minimizing the 2D reprojection error. Then, measured state variables are estimated through the location difference between the optimized results and the prior estimated ones. At last, EKF combines the prior estimation of state variables and the measured state variables according to the covariance matrix and outputs the post estimation of the state variables, where pose and position represent the trajectory. Each floating-point function circuit is assigned a priority level to ensure that each floating-point computation circuit can be accessed by only one floating-point function circuit at the same clock.

### 2.2. IMU Pre-Integration

IMU can measure acceleration and angular velocity through the accelerometer and gyroscope. Through integration, the rotation and displacement between two frames of images can be obtained. That is to say, if the position, velocity, and rotation at time *k* are known, these values at time *k* + 1 can be obtained. However, in the optimization algorithm, these values at each moment are estimated. When optimizing them, the data between two moments must be re-integrated, which makes the calculation requirements large. Lupton T. et al. [30] proposed IMU pre-integration to avoid repeated integration, and Forster, C. [31] made it better.

A schematic diagram of the IMU model is shown in Figure 3, and its formula is shown in (1). The measured values of linear acceleration and angular velocity are represented by $\hat{\left(\cdot \right)}$. Linear acceleration is the resultant vector of gravitational acceleration and object acceleration, $b_{a_{t}}$ and $b_{\omega_{t}}$ are offsets, $n_{a}$ and $n_{\omega }$ are Gaussian noise.
(1)$${\hat{a}}_{t}=a_{t}+b_{a_{t}}+R_{w}^{t}g^{w}+n_{a},\;{\hat{\omega }}_{t}=\omega_{t}+b_{\omega_{t}}+n_{\omega }$$

The representation of the position, velocity, and rotation (quaternion form) of k+1 frame can be obtained from k frame, as shown in (2). Here, b represents the body coordinate system, w represents the world coordinate system, $R_{t}^{w}$ represents the rotation matrix from the body coordinate system to the world coordinate system at time t, $q_{t}^{b_{k}}$ represents the rotation of the body coordinate system at time t relative to the body coordinate system at the time $b_{k}$, and $q_{b_{k}}^{w}$ represents the rotation of the body coordinate system relative to the world coordinate system at the time $b_{k}$.
(2)$$p_{b_{k+1}}^{w}=p_{b_{k}}^{w}+\nu_{b_{k}}^{w}\Delta t_{k}+\iint_{t\in \left[k,k+1\right]}\left[R_{t}^{w}\left({\hat{a}}_{t}-b_{\alpha_{t}}\right)-g^{w}\right]dt^{2},\;v_{b_{k+1}}^{w}=v_{b_{k}}^{w}+\int_{t\in \left[k,k+1\right]}\left[R_{t}^{w}\left({\hat{a}}_{t}-b_{\alpha_{t}}\right)-g^{w}\right]dt,\;q_{b_{k+1}}^{w}=q_{b_{k}}^{w}⊗\int_{t\in \left[k,k+1\right]}\frac{1}{2}\Omega \left({\hat{\omega }}_{t}-b_{\omega_{t}}\right)q_{t}^{b_{k}}dt$$
where
(3)$$\Omega \left(\omega \right)=\left[\begin{array}{cc} -{\left\lfloor w\right\rfloor }_{\times } & \omega \\ -\omega^{T} & 0 \end{array}\right],\;{\left\lfloor w\right\rfloor }_{\times }=[\begin{array}{ccc} 0 & -\omega_{z} & \omega_{y}\\ \omega_{z} & 0 & -\omega_{x}\\ -\omega_{y} & \omega_{x} & 0 \end{array}].$$

In (2), PVQ of the body coordinate system at time k+1 depends on $b_{k}$. If the PVQ of the body coordinate system is directly used as a variable to optimize and iteratively update, it will lead to a large amount of calculation. The idea of IMU pre-integration is to adjust the reference coordinate system from the world coordinate system w to the body coordinate system $b_{k}$ of k frame so that the integration result becomes the relative change of $b_{k+1}$ to $b_{k}$. It is realized by multiplying $R_{w}^{b_{k}}$, as shown in (4). The detailed derivation process is omitted [32].
(4)$$R_{w}^{b_{k}}p_{b_{k+1}}^{w}=R_{w}^{b_{k}}(p_{b_{k}}^{w}+\nu_{b_{k}}^{w}\Delta t_{k}-\frac{1}{2}g^{w}\Delta t_{k}^{2})+\alpha_{b_{k+1}}^{b_{k}},\;R_{w}^{b_{k}}v_{b_{k+1}}^{w}=R_{w}^{b_{k}}(v_{b_{k}}^{w}-g^{w}\Delta t_{k})+\beta_{b_{k+1}}^{b_{k}},\;q_{w}^{b_{k}}⊗q_{b_{k+1}}^{w}=\gamma_{b_{k+1}}^{b_{k}}.\;\alpha_{b_{k+1}}^{b_{k}}=\iint_{t\in \left[k,k+1\right]}\left[R_{t}^{b_{k}}\left({\hat{a}}_{t}-b_{\alpha_{t}}\right)\right]dt^{2},\;\beta_{b_{k+1}}^{b_{k}}=\int_{t\in \left[k,k+1\right]}\left[R_{t}^{b_{k}}\left({\hat{a}}_{t}-b_{\alpha_{t}}\right)\right]dt,\;\gamma_{b_{k+1}}^{b_{k}}=\int_{t\in \left[k,k+1\right]}\frac{1}{2}\Omega \left({\hat{\omega }}_{t}-b_{\omega_{t}}\right)\gamma_{t}^{b_{k}}dt$$

### 2.3. Algorithm of Feature Extraction

#### 2.3.1. FAST-9 Feature Detection

In the feature extraction technology, in addition to FAST [7], there are SIFT [5], SURF [6], and other algorithms. The features they extracted have strong invariance, but the time consumption is relatively large. In a system, feature extraction is only a part, and subsequent algorithms such as registration, purification, and fusion are also performed. This makes the real-time performance not suitable and reduces the system performance. The Fast detector is very fast because it does not involve complex operations such as scale and gradient. It uses the gray value of the pixel in a specific neighborhood to compare the size with the center point to determine whether it is a corner point.

The proposed accelerated core adapts the FAST-9 feature extraction [7] in the vision pipeline, a trade-off between accuracy and efficiency since the proposed accelerated core targets high-performance and low-power design.
(5)$$S_{c\to p}\{\begin{array}{c} darker,P<C-T\\ similar,C-T\leq P\leq C+T\\ brighter,C+T<P \end{array}$$

Here, *S* stands for the intensity relationship of the pixels *P* with the center *C*, and *T* is the user-defined threshold. If there are consecutive nine pixels darker or brighter, the center pixel is a FAST feature in the accelerated core.

This paper uses pipeline stages for hardware implementation since the FAST algorithm requires counters, which should be accumulated by clock cycles. However, it is not enough to implement it in 16 clock cycles, as shown in Figure 4, because a true circle path to access all pixels in the FAST algorithm should be half a circle. For example, if pixels P_1_, P_2,_ and P_10_–P_16_ are brighter while others are not, the center pixel should be a FAST feature. However, the first round only considers seven brighter pixels (P_10_–P_16_) which makes an incorrect judgment. The one-and-a-half circles for detecting solves this problem by redundantly calculating eight more pixels.

#### 2.3.2. Gradient-Based Score Calculation

This paper adapts a gradient-based score (Grad score in short) to indicate the quality of a feature after FAST feature detection, as shown below:(6)$$Grad\;score=\overset{6}{\underset{i=1}{\sum }}\overset{6}{\underset{j=1}{\sum }}{\left(P_{i,j+1}-P_{i,j-1}\right)}^{2}/64+{\left(P_{i+1,j}-P_{i-1,j}\right)}^{2}/64$$

Here, the input patch is size 8 × 8, and *P* stands for the pixel values. Therefore, the Grad score is calculated with a 6 × 6 window inside the patch. The score will be used in the rest process in the core.

### 2.4. Algorithm of Feature Coordinates Alignment

Feature coordinates calculated from IMU pre-integration and feature prediction are corrected by photometric error, which is based on the difference in pixel data between the old feature patches in the last frame and the new feature patches extracted in the current frame. The photometric error is defined by:(7)$$e_{l,j}=P_{l}\left(p_{j}\right)-I_{l}\left(ps_{l}+Wp_{j}\right)-m$$
where the scalar factor $s_{l}=0.5^{l}$ stands for the down-sampling number of layers of the image pyramid. W stands for the warping transform matrix in [33], $p_{j}$ is the patch pixel in the patch $P_{l}$, and p is the feature coordinate.

This paper adapts the QR-decomposition method in [33], which stacks all error terms together for given estimated coordinates $\hat{p}$:(8)$$b\left(\hat{p}\right)=\bar{A}\left(\hat{p}\right)\delta p$$

Here, $\bar{A}\left(\hat{p}\right)$ is based on the patch intensity gradients along the X and Y axes. The QR-decomposition of $\bar{A}\left(\hat{p}\right)$ obtains an equivalent reduced linear equation system:(9)$$b\left(\hat{p}\right)=A\left(\hat{p}\right)\delta p$$

The iteration will be user-defined and ten times in the proposed core. If the photometric error is still higher than a certain threshold, the feature will be marked as bad and abandoned, while others could be used in the EKF update.

### 2.5. Supported Operations in SLAM Algorithms

Most SLAM algorithms require not only basic operations, such as the addition or multiplication of scalars, but also operations of rotation dynamics, including rotation matrix, quaternion, and lie algebra. The proposed VIO accelerated core provides full functionality to support the abovementioned operations, shown in Table 1.

## 3. Hardware Design

This section introduces the overall hardware architecture of the proposed VIO accelerated core and details four important submodules and techniques adapted in the design.

### 3.1. Overall Hardware Architecture

Figure 5 shows the proposed accelerator’s overall architecture constructed by four sub-modules, including a fixed-point vision pipeline, a memory interface, a programmable computation core, and layers for the EKF engine.

The vision pipeline receives input data from the image sensor to perform new and predicted feature extraction. The memory interface uses a shared memory strategy to store intermediate values and output data. The programmable computation core pre-loads three programs to perform the computation of IMU pre-integration and the computation of the Jacobian matrix. This core can satisfy the pose estimation of a sample class of Kalman-filter-based SLAM algorithms. In the EKF engine, layer 1 contains a finite state machine in charge of the whole process. The layer of the feature processing engine scores and sorts the features received from the feature extraction engine then compares the best 25 of them with the features stored in the feature manager and replaces the old ones with them if the new features are better. The feature processing engine also computes the 2D location difference of the same feature in the last and current frames. Then, the EKF engine completes the update process of the EKF process with the help of a specific mission layer and FP arithmetic. The specific mission layer shares the floating-point arithmetic with the computation core for higher hardware utilization.

### 3.2. Fixed-Point Vision Pipeline

Figure 6 details the structure of the fixed-point vision pipeline, including a Gaussian pyramid-image half sample engine, a FAST feature identifier, a Grad score-based patch extractor, and a patch controller. 

The architecture workflow is pipelined with line buffers, thus reducing memory consumption and increasing processing throughput. The operation provides a foundation for the following process by interpreting images at multiple resolutions and different scales with a Gaussian sampling pyramid of the input grayscale image. The half-sampled pixels are transferred through an asynchronous FIFO, followed by a group of 11 line buffers. The rest processes are based on an area-reused 12 × 12 sliding window where pixel data can be available for three layers.

The FAST identifier is a feature extractor adapting the FAST-16 algorithm where if a pixel differs greatly from enough pixels in its surrounding neighborhood, the pixel may be a corner. Adapting the FAST algorithm for every pixel indicates a continually analyzed 16 pixels, i.e., whether the pixel is brighter/darker than the center pixel as the last pixel does. In the hardware implementation, the algorithm is processed in a pipeline which takes 24 periods. At every stage, the circuit will detect the brightness for both the current and last pixels and increase or reset the counters. In Figure 6, BrightCnt means the number of continuing pixels with a grey value more significant than the grey value of the center plus threshold, while DarkCnt stands for the opposite. If BrightCnt or DarkCnt is larger than 9, the pixel is identified as a FAST feature, and the result will be passed to the patch controller by a line buffer.

The FAST identifier is optional, considering the patch extractor has already been attached with a Grad score comparator which adapts only the front-computing. Input pixel data from the 8 × 8 window generates their gradient values which are used to calculate the Grad score according to (6).

The patch controller temperately stores the information of a maximum of 100 patches in sub-patch controllers, where a patch is defined by the feature pixel and its surrounding 12 × 12 window of pixels. In order to avoid the extracted feature points being too concentrated, this paper proposed a patch extract strategy, which constructs banned rows and banned columns. When a new feature/patch is extracted, its surrounding 16 rows and columns will be banned for the subsequent extraction unless the new feature has a higher Grad score. The 100 sub-patch controllers are awakened one by one to reduce power consumption and send write to enable signal to the patch controller for patch storing, which is implemented by dual-port rams. If the maximum feature limit (100) is reached, a special sub-patch controller will be used to cover the patch with the least Grad score if the new feature has a higher score, ensuring the extracted patches are high quality for the rest process. At the end of a frame, the vision pipeline sends a signal to the top state machine, indicating the available feature patches.

### 3.3. Programmable Computation Core

Figure 7 details the structure of the programmable computation core realized by instruction control, which is an extension of the previous work [34]. A matrix computation core performs the matrix operation and controls the program workflow, and a special computation core performs quaternion, angle-axis, and scalar operations. The computation core is controlled by a proposed instruction set architecture (ISA). Operation instructions store the offset address of operands, source, and target address, transpose or generate antisymmetric matrix, and the operation types. The PC reads instructions from IRMEM and sends them to the corresponding instruction decoder for operation instructions. Each core has its own 38-bit instruction set and an 8-bit address instruction memory but shares a 4 × 7-bit address, 32-bit per data, memory for operators. Each core has and only has access to its own nine adders, nine multipliers, one floating sqrt, and one reciprocal computation circuit. Compared with the matrix core, the special core has some unique computation circuits, such as cosine and sine. Two instructions for matrix core, bubble and sync, and one for special core, bubble, are used to synchronize the cores and secure the computation order. An interface and matched instructions for 16-bit per address, 38-bit per data memory are kept for the potential memory. The operations required by different Kalman-filter-based algorithms can be programmed into the instruction and operator memory blocks. What is more, Table 2 shows the supported operations by the computation core, including normal scalar operations and special operations required by most SLAM algorithms.

### 3.4. Feature Processing Engine

Figure 8 shows the structure of the feature processing engine, which includes a memory interface, a feature manager, a floating-point patch workspace, a coordinates alignment circuit, and interaction with the EKF engine. During the process of VIO, after IMU pre-integration, the pose of the feature concerning the last frame is exported out of the operator memory.

The patch workspace requests the 12 × 12 feature patch in a fixed-point format whose center locates where the feature is in the last frame from the feature extraction engine in the fixed-point vision pipeline, converts it to the floating-point format, and stores it temporarily into the dual-port rams. A bilinear interpolation is performed to sample an 8 × 8 patch from the original 12 × 12 patches according to the warping matrix and the updated patch coordinates [33]. Then, the patch workspace measures the difference between the old and the new patches, which are the gray value difference and grad difference along the X and Y axes. The information will be output to the coordinate alignment circuit. 

The coordinates alignment circuit generates matrix A and vector b according to (7) and (9). The least-square equation optimizes the feature location and output to patch the workspace for resampling. The process ends when the iteration reaches the time limit or the 2D reprojection error is lower than a specific threshold. If the reprojection error is acceptable, the feature processing engine computes and stores the Jacobian matrix of measured results and noise in memory. EKF engine performs the update of the Kalman filter. After finishing the update, the feature processing engine sorts and compares the newly extracted features with the old ones. The feature processing engine replaces the old feature information and patches with the new one if the new feature is better. 

In order to save resources and reduce power consumption, 3 × 3 operation cores are used for matrix-vector operations of any size. To this end, this paper innovatively adopts the idea of vectorization, which is in matrix multiplication, the left matrix is input to the column from left to right, and the right matrix is input to the row from top to bottom. Each set of operations completes an M33 matrix, calculates a new M33 matrix every three rows and three columns, as shown in Figure 9, and finally, calculates the entire large matrix with low resource and power consumption. 

## 4. Implementation Results

### 4.1. Accuracy of Datasets Compared with Software Platform

A comparison to the software platform on a dataset is performed to verify the functionality and accuracy of the proposed VIO accelerator. The following were the main parameters of the PC: Ubuntu 18.04 (64-bit), Intel(R) Core (TM) i5-8265U CPU with 8G RAM. Moreover, we translate the ROVIO [33] algorithm into the ISA proposed by this paper. The dataset used in the evaluation is EuRoC, which is one of the most authoritative and challenging datasets. We translate the EKF process of ROVIO into the proposed ISA and evaluate them on EuRoC. As shown in Table 3, the accuracy loss is neglectable, indicating the high accuracy of the proposed accelerated core.

### 4.2. Evaluation Platform and Experiments

The evaluation platform is constructed by a four-wheeled platform that can move smoothly, carrying the Xilinx XCVU 440 FPGA evaluation board with monocular camera mt9v034 and MPU9250, as shown in Figure 10. Figure 11 illustrates the measured experience map over frames during a typical SLAM operation in a flat corridor, which is about 70 m in total. The evaluation platform starts from one of the corners and moves smoothly at a speed of 0.35 m/s on average. After a loop of testing, we find that the trajectory is only slightly shifted, indicating the robust functionality and relatively high accuracy of the proposed accelerated core.

### 4.3. Discussions

Table 4 illustrates the implementation results of the proposed reconfigurable VIO accelerator. ASIC synthesis is performed to better compare with state-of-the-art ASIC implementation results. This work operates in the highest frequency both in FPGA and ASIC synthesis. Nevertheless, the on-chip memory usage is the lowest among these works for occupying only 70 KB when [29] is lower but with the usage of build-in SoC. Compared with FPGA implementations, slice LUTs and FFs consumed by the accelerator are 2.05× and 2.35× less than [28]. DSP consumption in this work is 8.03× and 1.8× less than [28] and [29]. As illustrated in Table 4, the core area in 28 nm CMOS technology is only 3 mm^2^, which is 6.67× and 3.64× smaller in comparison to [27] and [26]. This work supports reconfigurable SLAM architecture, while others do not.

The power consumption on the FPGA platform is about 0.683 W, which is lower than the 1.46 W on the Kintex-7 XC7K355T [28]. The power proportion of each module is shown in Figure 12. Based on the synthesis results in the 28 nm CMOS process, the power consumption is only 179.52 mW, which is much better than that in [26] with the same 28 nm CMOS technology, but still worse than that in [27] with 65 nm CMOS technology.

Due to the lightweight characteristic of the EKF optimization and FAST-based features, the proposed accelerated core essentially reduces the consumption of LUTs, FFs, and DSPs. Secondly, compared with other backend optimization techniques, such as bundle adjustment, EKF does not require storing the entire image, significantly reducing a large amount of on-chip memory. Finally, the delicate design improves the peak performance of the proposed accelerated core.

## 5. Conclusions

This paper proposed a reconfigurable visual inertial odometer (VIO) for the simultaneous localization and mapping (SLAM) application in autonomous mobile robots. The proposed accelerated core leveraged a lightweight feature extraction algorithm with the extended Kalman filter (EKF) update algorithm to provide high energy efficiency while achieving appropriate SLAM results. Furthermore, the reconfigurability of the accelerated core brings more potential downstream exploration for AMR applications. 

The main hardware architecture of the VIO accelerator consisted of four sub-modules: (1) fixed-point vision pipeline, (2) memory interface, (3) programmable computation core, and (4) layers for the EKF engine. We first evaluated the accuracy both on benchmark datasets and real experimental tests. As shown in the implementation results, the on-chip memory usage of 70 KB was the lowest among the standalone works for SLAM. Meanwhile, the hardware-resource usage and power dissipation on the FPGA implementation and the synthesis of 28 nm CMOS technology also outperformed the state-of-the-art works on the same condition.

## Figures and Tables

**Figure 1 sensors-22-07669-f001:**
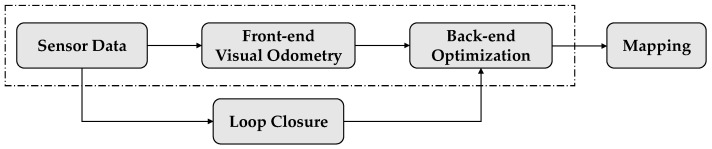
Process Diagram of SLAM.

**Figure 2 sensors-22-07669-f002:**
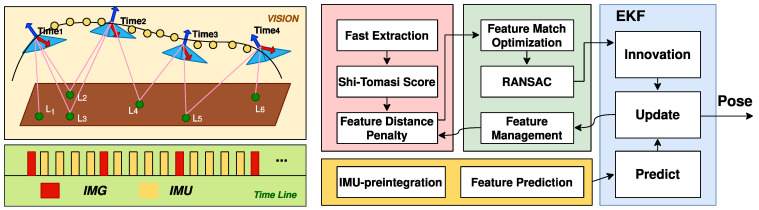
The overall procedure of the visual–inertial odometry.

**Figure 3 sensors-22-07669-f003:**
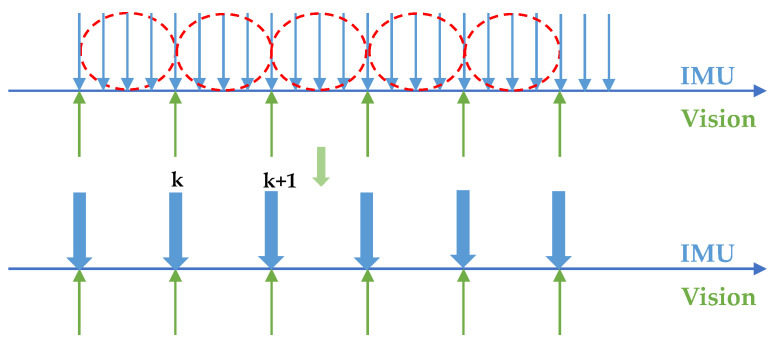
Diagram of IMU model.

**Figure 4 sensors-22-07669-f004:**
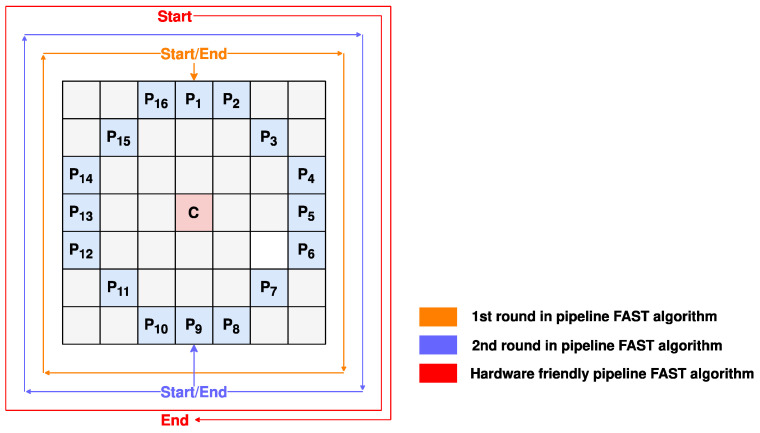
The hardware-friendly pipeline FAST algorithm.

**Figure 5 sensors-22-07669-f005:**
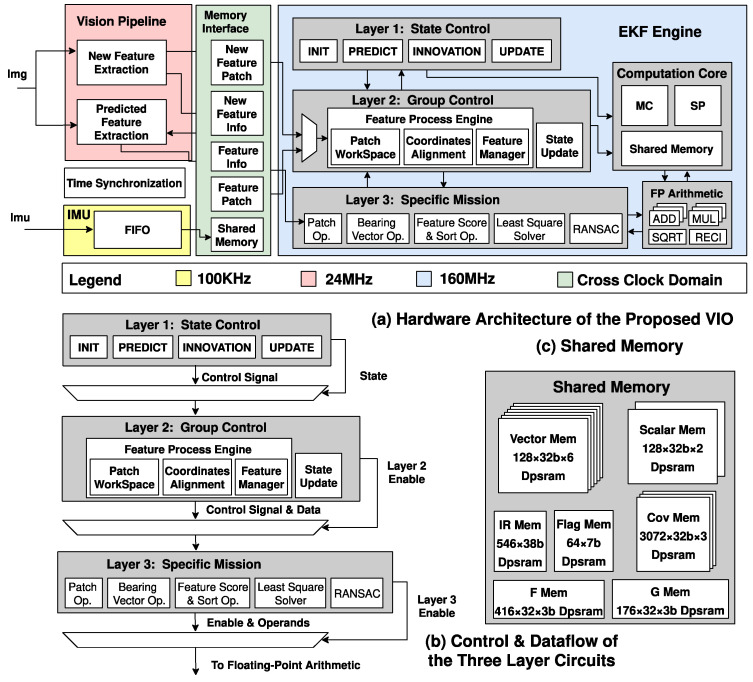
Overall hardware architecture of the proposed accelerated core. (**a**) Hardware architecture of the proposed VIO accelerated core; (**b**) control and dataflow of the three-layer circuits; (**c**) shared memory for the storage of vectors and matrices.

**Figure 6 sensors-22-07669-f006:**
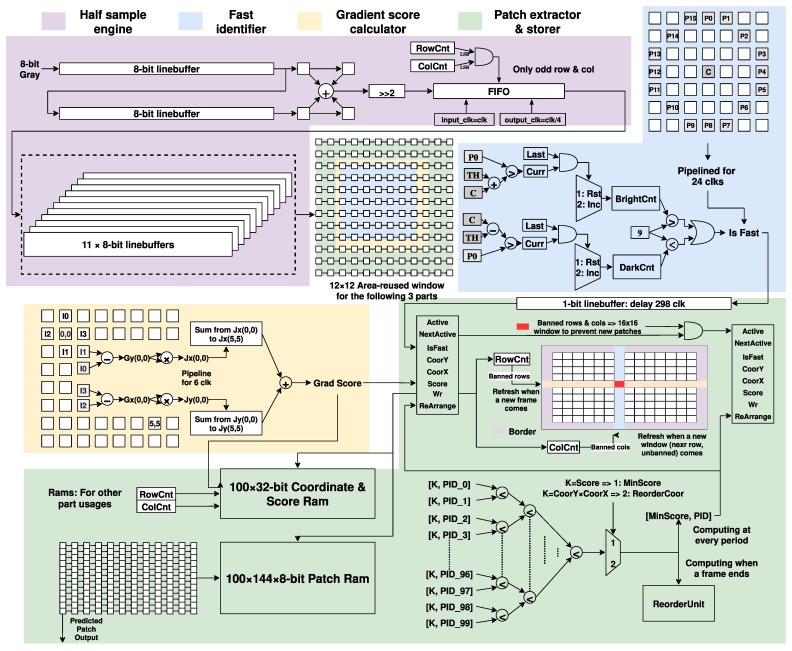
Detailed structure of the fixed-point vision pipeline.

**Figure 7 sensors-22-07669-f007:**
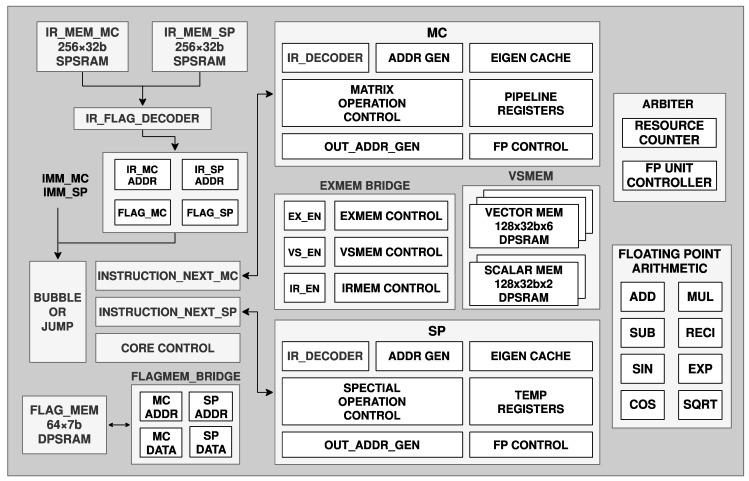
Modules and processing flow of the programmable computation core.

**Figure 8 sensors-22-07669-f008:**
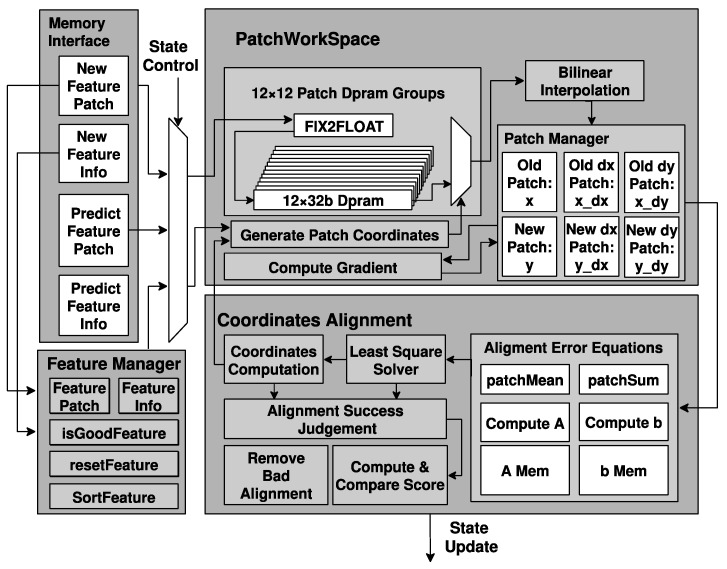
Detailed structure of the feature processing engine.

**Figure 9 sensors-22-07669-f009:**
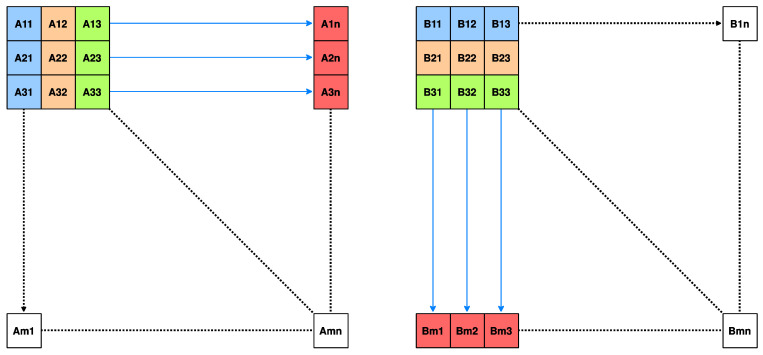
Schematic diagram of vectorized matrix multiplication strategy.

**Figure 10 sensors-22-07669-f010:**
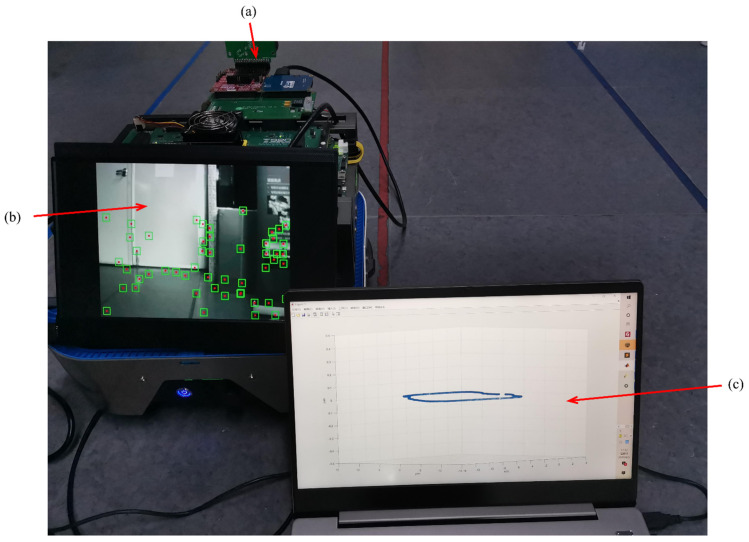
Demo evaluation platform. (**a**) VU440 FPGA board with mt9v034 image sensor and MPU9250 IMU. (**b**) New features are shown through the HDMI display. (**c**) Trajectory output in the x, y, and z axes (three dimensions).

**Figure 11 sensors-22-07669-f011:**
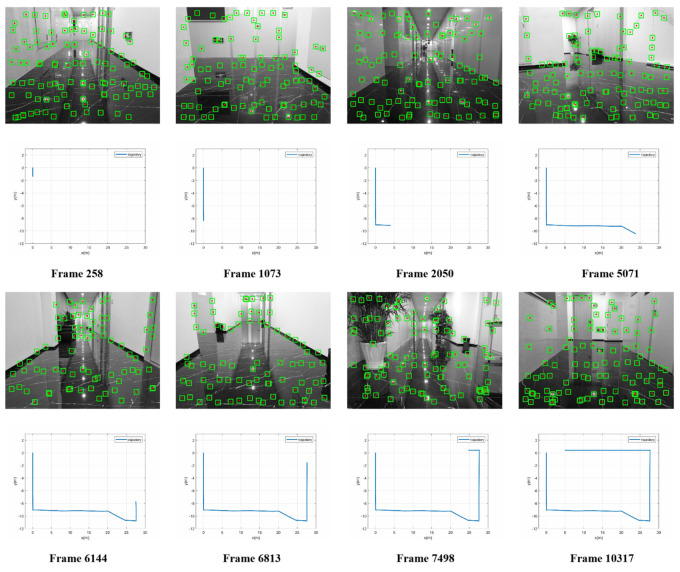
Measured experience map over frames during SLAM process.

**Figure 12 sensors-22-07669-f012:**
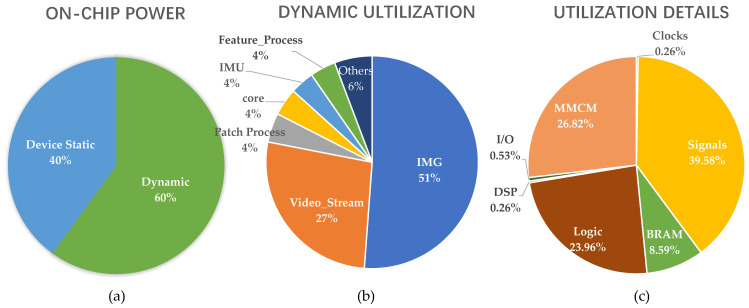
Power proportion of each module. (**a**) on-chip power; (**b**) dynamic utilization; (**c**) utilization details.

**Table 1 sensors-22-07669-t001:** Supported operations by the accelerated core.

Operation	Description	Time Consumption
Scalar add/sub	Scalar addition/subtraction	5 clock cycles
Scalar mul	Scalar multiplication	2 clock cycles
Scalar reci	Scalar reciprocal	10 clock cycles
Scalar sqrt_slow	Scalar square root with high accuracy	27 clock cycles
Scalar sqrt_fast	Scalar square root with low latency	10 clock cycles
Sin_Cos	Sine and cosine function for input in radians	52 clock cycles
M_inv	Matrix inversion	31 clock cycles
Li2R	Transform lie algebra to a rotation matrix	73 clock cycles
Li2Q	Transform lie algebra to quaternion	65 clock cycles
R2Q	Transform rotation matrix to quaternion	53 clock cycles
Q2R	Transform quaternion to a rotation matrix	14 clock cycles
Q_q	Quaternion multiplication	14 clock cycles

**Table 2 sensors-22-07669-t002:** Description and time consumption of operations supported by the computation core.

Operation	Description	Time Consumption
Scalar add/sub	Scalar addition/subtraction	5 clock cycles
Scalar mul	Scalar multiplication	2 clock cycles
Scalar reci	Scalar reciprocal	10 clock cycles
Scalar sqrt_slow	Scalar square root with high accuracy	27 clock cycles
Scalar sqrt_fast	Scalar square root with low latency	10 clock cycles
Sin_Cos	Sine and cosine function for input in radians	52 clock cycles
M_inv	Matrix inversion	31 clock cycles
Li2R	Transform lie algebra to a rotation matrix	73 clock cycles
Li2Q	Transform lie algebra to quaternion	65 clock cycles
R2Q	Transform rotation matrix to quaternion	53 clock cycles
Q2R	Transform quaternion to a rotation matrix	14 clock cycles
Q_q	Quaternion multiplication	14 clock cycles

**Table 3 sensors-22-07669-t003:** Evaluation of accuracy.

Dataset	ROVIO (Software)	The Proposed Core
MH_1	0.19%	0.19%
MH_2	0.23%	0.23%
MH_3	0.47%	0.49%
MH_4	0.55%	0.52%
MH_5	0.78%	0.79%
V1_1	0.28%	0.26%
V1_2	0.35%	0.34%
V1_3	0.27%	0.25%
V2_1	0.26%	0.26%
V2_2	0.37%	0.40%
V2_3	0.61%	0.61%

**Table 4 sensors-22-07669-t004:** Hardware implementation results. The outstanding work in the comparison is bold.

	MIT 2017 [28]	ICFPT 2021 [29]	JSSC 2019 [27]	ISSCC 2019 [26]	This Work
Type	VIO	SLAM	VIO	SLAM	VIO
Odometry	IMU	Visual	IMU	Visual	IMU
FPGA Platform	Kintex-7 XC7K355T	UltraScale + XCZU7EV	N/A	N/A	UltraScale + XCVU440
Technology	N/A	N/A	65 nm	28 nm	28 nm *
Resolution	N/A	640 × 480	752 × 480	640 × 480	640 × 480
Speed	20 fps	15.5 fps	**171 fps**	80 fps	160 fps
Frequency	100 MHz	100 MHz	62.5 MHz/ 83.3 MHz	240 MHz	**160 MHz** **(250 MHz *)**
SoC	No	Yes	No	No	No
On-chip Memory	2048 KB	**61 KB**	854 KB	1126 KB	70 KB
LUTs	192,000	146,572	N/A	N/A	**91,802**
FFs	144,000	74,166	N/A	N/A	**61,107**
DSPs	771	173	N/A	N/A	**96**
Power	1.46 W	Not Given	24 mW	243.6 mW	**0.683 W** (179.52 mW)
Area	N/A	N/A	20 mm^2^	10.92 mm^2^	**3 mm^2^**
Reconfigurable	No	No	No	No	Yes
Application	Nano and pico robots	Autonomous navigation	AR, VR and UAVs		AMRs

*: Synthesis results in 28 nm CMOS technology.

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
