# Peer review of "A Reconfigurable Visual–Inertial Odometry Accelerated Core with High Area and Energy Efficiency for Autonomous Mobile Robots"

_sensors, 2022, doi:10.3390/s22197669_

Round 1

Reviewer 1 Report

This paper introduces a reconfigurable, real-time, low-area, and energy-efficient VIO odometry accelerator implemented by FPGA. There are three main innovations: (1) a reconfigurable accelerator architecture that adapts to different VIO algorithms, (2) an optimized instruction-based structure supporting the simultaneous workflow of fixed- point and floating-point units to accelerate VIO algorithms, (3) a computing core with shared memory and processing element to reduce area and power consumption. The paper is well-written and structured. The reference is up-to-date. Below are several minor comments:

1)   The work proposes a programmable computation Core for EDF algorithm. It's better to introduce how the compiler is designed.

2)   The work proposes a reconfigurable architecture. However, it is not explained how the architecture is reconfigurable and whether it supports other backend optimization algorithms for SLAM.

3)It is better to show the contribution of each innovation point to the high performance result.

4) Is it possible add some analysis about power consumption, which may be interesting to the readers.

5)   Why only five sequences of the EuRoc are shown? It is better to show the results of all the sequences.

6)   Some sentences need to be polished. For example, line 232, “If BrightCnt or DarkCnt. This is larger than 9, the pixel is identified as a FAST feature…”

Author Response

We are grateful for your effort to improve the manuscript. Please see the attachment for our responses and revisions. Thank you in advance.

Reviewer 2 Report

This work developed a highly integrated system for positioning autonomous mobile robots (AMRs). It claims several advances of this work, including the reconfigurable architecture, hardware-oriented algorithm optimization, high positioning area, energy efficiency, high precision, etc. However, publishing this work should only be considered if the following concerns have been resolved.

1. please provide more labels in figure 10;

2. please provide more text explaining the experiment, like the path of AMR, for 4.2.2.

3. please highlight the advantages of this work in table 4 over other techniques and the explanation of why and how to reduce these consumptions.

4. please check and correct a few typos.

Author Response

(The authors gave the same response as above.)

Reviewer 3 Report

I have examined the manuscript number: sensors-1924431 “A Reconfigurable Visual-Inertial Odometry Accelerated Core with High Area and Energy Efficiency for Autonomous Mobile Robots". In my opinion this article is well written and in general is of great importance in the field of Sensing and Imaging.

·        In abstract, major data obtained from the results should be added.

·        Discussion section should be elaborated comparing with the previous studies.

·        In conclusion section, major data obtained from the results should be added.

Author Response

(The authors gave the same response as above.)

Round 2

Reviewer 2 Report

all my comments are well addressed.